# Assessment of Physicians’ Willingness to Work with Patients Not Yet Diagnosed with COVID-19 in a Romanian Sample

**DOI:** 10.3390/healthcare12020161

**Published:** 2024-01-10

**Authors:** Tudor-Ștefan Rotaru, Daniela Cojocaru, Ștefan Cojocaru, Ovidiu Alexinschi, Aida Puia, Liviu Oprea

**Affiliations:** 1Department of Bioethics, University of Medicine and Pharmacy “Gr. T. Popa” Iași, 700115 Iași, Romania; tudor.rotaru@umfiasi.ro (T.-Ș.R.); liviu.oprea@umfiasi.ro (L.O.); 2Department of Sociology and Social Work, University “Alexandru Ioan Cuza” of Iași, 700506 Iași, Romania; danac@uaic.ro (D.C.); contact@stefancojocaru.ro (Ș.C.); 3Department IIIA, “Socola” Institute of Psychiatry, 700282 Iași, Romania; 4Department of Community Medicine, “Iuliu Hațieganu” University of Medicine and Pharmacy, 400347 Cluj-Napoca, Romania; aida.puia@umfcluj.ro

**Keywords:** physicians’ perceptions, willingness to work, COVID-19, personal protective equipment

## Abstract

Background: The risk to physicians who worked with patients without confirmed COVID-19 testing during the pandemic has been little studied. However, they were at high risk. Methods: In the summer of 2020, 1285 Romanian physicians participated in a single-center study. Participants (mean age = 48.21 years; 302 males and 982 females, all specialties) completed a series of single-item measures adapted from previous studies on work ethics and responsibility. This study used Mann–Whitney comparisons between physicians who reported that they knowingly had direct contact with COVID patients and those who did not regarding their willingness to work. Results: Compared with their colleagues, physicians who reported not knowingly having direct contact with COVID patients reported less access to protective equipment, less overall willingness to respond when asked to work with infected patients, more likely to work out of fear of losing their jobs, and fear of legal repercussions. They received less training in the use of protective equipment. Conclusions: Physicians who worked with patients not yet diagnosed with COVID-19 were significantly less willing to work. The perception of invisible risk may explain the observed differences.

## 1. Introduction

Understanding the challenges to health workers’ willingness to work in pandemic settings in the past is critical to predicting and improving their engagement in the future. In a 2020 Australian survey of 580 health professionals, including doctors, nurses, and paramedics, 75% faced a dilemma between professional duty and personal risk during the COVID-19 pandemic. The study focused on their willingness to work amid potential contagion and transmission of the virus to family members. The results showed differences in risk acceptance, with nurses more likely to accept inherent risks than physicians. Critical factors influencing willingness to work were the availability of personal protective equipment (PPE), family safety concerns, and the risk of COVID-19 exposure. The role of PPE was paramount, overshadowing other factors in decision making, with its impact on willingness to work second only to combined family concerns and risk of exposure [1].

In a 2020 study of 387 medical professionals, 74.3% of 187 physicians continued to work during COVID-19, with 66.8% feeling ethically obligated to do so. Factors influencing their willingness included personal health (57.2%), skills (34.8%), and availability of PPE (51.3%). Incentives such as hazard pay, vaccination guarantees, and medical coverage were important motivators. Among 200 medical students, responses were mixed. More than half feared contracting COVID-19 and transmitting it to family members. Confidence in the effectiveness of PPE varied, with the highest confidence in N95 masks (35.3%) [2]. A Pakistani study of 250 medical professionals using a 20-item tool showed a 42.6% willingness to work during COVID-19. Factors influencing willingness included perceived threat, role competence, self-efficacy, and duty, but individual risk perception and role importance had less impact [3].

A 2020 study in Bangladesh surveyed 422 doctors during the COVID-19 pandemic. A total of 96% understood the health implications, 85% felt knowledgeable about protective measures, but half found PPE inconvenient to use. Willingness to work correlated with understanding COVID-19 risks, PPE effectiveness, and lower risk perception. A total of 69.7% were willing, 8.9% were unwilling, and 21.4% were ambivalent. Younger physicians and those with previous pandemic experience were more willing. Factors that decreased willingness included seniority, direct contact with COVID-19, family concerns, and personal comorbidities. Overall, perception of protective equipment, understanding of safety measures, and lower perceived risk of workplace infection influenced willingness to work [4].

A study in Palestine assessed the willingness of HCWs to serve during COVID-19 using surveys and interviews. Of 357 valid responses, 24.9% were reluctant to serve, with primary healthcare workers being more willing. HCWs with children and less concern about contracting the virus showed greater willingness. Willingness was associated with perceived vulnerability, COVID-19 severity, and experience with pandemics. Stress and disillusionment decreased willingness. The study showed that HCWs’ risk perceptions, professional obligations, and challenges influenced their service commitment during the pandemic [5].

Availability of PPE, fear of family transmission, ethical duty, financial incentives, early vaccination, and training influence health workers’ willingness to work [1,2,3,4,5]. All of the studies we found examined the relationship between willingness to work and various factors, but they did not take into account that physicians who did not knowingly work with cases diagnosed with COVID-19 were also at risk. These risks may have influenced physicians’ willingness to work even when the patient they were working with was not explicitly diagnosed with COVID-19. The purpose of this study was to make a descriptive comparison between physicians who knowingly worked with patients infected with COVID-19 and physicians who did not knowingly work with patients infected with COVID-19 in terms of their willingness to work with or without PPE, fear of losing their job if they refused to work, access to PPE, and PPE training.

## 2. Materials and Methods

### 2.1. Participant Recruitment and Study Design

The purpose of this study was to make a descriptive comparison between physicians who knowingly worked with patients infected with COVID-19 and physicians who were not known to have knowingly worked with patients infected with COVID-19 in terms of their willingness to work with or without PPE, fear of losing their job if they refused to work, access to PPE, and PPE training. The data collected in this research are part of a larger study of pandemic work ethic, responsibility, and willingness to work.

Physicians were recruited to participate in the study through an email sent as a newsletter directly to their email addresses (Figure 1). Access to the physicians’ email addresses was provided by the Romanian National College of Physicians and the territorial colleges of physicians. The newsletter announcing participation in the study was sent three times to each physician. The data collection was carried out between 1 July and 31 August 2020. Regarding the inclusion criterion, all physicians who had a valid license to practice in Romania at the time of data collection could participate in the study. The Romanian College of Physicians is the national professional organization of physicians, an institution serving the public interest, and is non-governmental, non-political, and non-profit. It is the national regulatory body for all physicians in Romania. When estimating the sample size, the aim was to cover as many physicians with a valid right to practice in Romania as possible. No calculations were made prior to the study to estimate the required sample size. This was a single-center study. Data were collected from across the country, but the database and statistical processing were performed using a single research team.

In terms of study design, we set out to use multiple items to measure several constructs found in the literature: a measure of duty, a measure of agency, a measure of perceived moral support, and a measure of willingness to work. The initial goal of the study was to observe whether the data in our study would correlate with each other according to theoretical models found in the literature (e.g., whether increased levels of agency would correspond to increased levels of work effort). One flaw in the study design that we noticed after data collection was the situation of items that did not apply to certain cases (e.g., in item 17, “I was willing to work even if my partner got COVID-19”, many participants answered “not applicable”). Therefore, we could not proceed with data processing according to the original design because we could not perform parametric statistics. The number of observations that had an answer other than “not applicable” to all questions in a construct remained very small. In the present study, the independent variable was the dichotomous measure: knowingly worked with cases diagnosed with COVID-19 versus not knowingly worked with cases diagnosed with COVID-19. The dependent measures were responses to items related to willingness to work, access to PPE, and other variables relevant to the literature to date. The present study had a purely descriptive design.

The study was conducted between July and August 2020. A total of 1285 Romanian physicians participated, addressing issues such as responsibility, medical ethics, professional drive, and self-confidence during the first wave of the pandemic. The respondents reflected a national spectrum, both demographically and geographically. Of these, 982 were female, 302 were male, and one physician preferred not to disclose his or her gender, reflecting the national gender distribution of physicians. The mean age was approximately 48 years. Given the convenience sampling approach, national representation was a goal but not a guarantee. Distribution was primarily web-based through local medical society chapters to achieve broad coverage. The exact participation rate remained uncertain due to the multi-channel distribution strategy.

### 2.2. Questionnaires

The full questionnaire was constructed from existing constructs in the literature, and the initial intention of the study was to test how participants’ responses were distributed according to theoretical models of responsibility and work willingness (Appendix A). Several studies published at the time of the questionnaire were used to formulate the items [6,7,8,9,10]. At the beginning of the online survey, participants were presented with an informed consent document to review and accept before proceeding. This consent document detailed the purpose of the study, the researchers involved, and the methods of data collection. Participants were informed: “The information you provide will not be linked to your identity […] Completing this survey may increase your awareness of the impact of the COVID-19 pandemic on your personal well-being and your professional experiences during its duration”. Regarding confidentiality, the segment clarified: “By agreeing to participate in this research, your legal rights will remain intact. The information you contribute to this investigation will be collected and secured under the General Data Protection Regulation (GDPR)”. The concluding segment of this consent portion read: “By selecting the CONTINUE option, I give my consent to participate in this research”.

Table 1 details the content of the questionnaire.

Question 19 focused on physicians’ self-reported illnesses. Because the online format allowed respondents to choose from a diverse list, each disease in our database was divided into a yes-or-no format (e.g., “Have you ever had bronchial asthma, including allergic variations?”). Splitting the choices into dichotomous formats was critical to interpreting the data from this question. Question 26 asked about the specific healthcare settings in which they worked, such as maternity wards or intensive care units. As participants could select multiple settings, these were also dichotomized. Question 99 asked physicians to rank certain factors that influenced their medical practice. Questions 129 and 130 similarly asked for sources of information and supporting resources. These rankings were then deconstructed into distinct variables. The following questions, from 38 to 144, used a 6-point Likert scale and explored various theoretical constructs. While these items were not grouped by their respective constructs in the database, each was labeled accordingly. After data collection, certain items were excluded due to their inapplicability to various physicians. Thus, the collected data were not consolidated, and internal consistency metrics were not calculated. IBM SPSS Statistics 20.0 was used for data processing.

Physicians also responded to statements using a six-point scale to indicate agreement or disagreement. This specific scale was designed to eliminate a midpoint, thus preventing neutral responses. The broader research was designed to examine responses in the context of theoretical relationships among self-confidence, work ethic, and professional commitment. However, data characteristics prevented linear regression due to the skewed distribution and prevalence of “not applicable” options. These options were critical to reflect realistic scenarios. An example was item 48: “When asked to work with COVID-19 patients, I was willing to respond”. Because not all physicians were asked to work with COVID-19 patients, there was a “not applicable” response option. Another example was item 53: “I was willing to provide direct patient care even though I did not have access to a K95 mask, although I should have used it”. Many physicians have had access to masks from the beginning and have never been faced with this situation where they had to decide whether to risk their health. Given these data limitations, non-parametric data treatment was deemed the most appropriate analytical approach.

In terms of validating the questionnaires on the study population, the items we included in the study were meant to be part of some constructs (some factors or dimensions) such as a sense of duty, perception of a threat, sense of trust in the institution, sense of trust in the government, etc. Because most of the items had a “not applicable” variant that we had to introduce, we had to forego aggregating the items into constructs (in separate questionnaires). We did not calculate a Cronbach’s alpha internal consistency index or a total score for the scales. This, unfortunately, resulted in an existing collection of single-item measures. However, where we were able to identify a criterion variable, we conducted binomial logistic regressions, which we have published elsewhere. For the present study, we were able to present a comparative analysis of the two types of physicians: those who knowingly worked with cases diagnosed with COVID-19 and those who did not knowingly work with such cases.

It is also important to specify that a logistic regression model would have been appropriate, especially since linear regression could not be used for the reasons stated above. However, the logic of the design would have required that the dichotomous variable used as a criterion in the logistic regression approach represented a temporally ordered phenomenon according to the aspects stated in the predictors. Since we were not able to claim that factors related to training and use of PPE in any way predicted the membership of doctors in the category of those who worked with explicitly diagnosed cases or not, it would have been illogical to use a regression model.

### 2.3. Statistics

The statistical test used in the study was the Kolmogorov–Smirnov test, which tested the difference between a given distribution and the theoretical normal distribution. A significant Kolmogorov–Smirnov test indicated that there was a significant difference between the observed data and the normal distribution. This meant that parametric statistical tests could not be used for these data. We also used the Chi-Square statistical test to show whether there was a difference between the observed frequency and the expected frequency along the categories formed by the meeting of two categorical variables (in our case, the gender of the research participants and whether they worked with diagnosed cases). A significant Chi-Square test showed that there was a significant difference between the observed and expected frequencies across the categories, which meant that the sexes were not evenly distributed across the two categories of doctors, which could have influenced the observed differences to some extent. Finally, the main statistical test used was the Mann–Whitney, which was suitable for testing whether there was a significant difference between two independent groups in the mean response to a given question asked on a scale. When the distribution of responses to this question was not normal, the Mann–Whitney test used a rank average. It is worth adding, in this section, that the use of non-parametric tests on non-normally distributed data was not only methodologically sound but also protected against false-positive results. SPSS 20.0 was used for statistical processing.

In our research, we also used the statistical calculation of Cohen’s d indicator for effect size. The reason for using this indicator was to have a more competent discussion of the results. Some effects were significant, but they were small. Other effects were larger and deserved more attention. In order to critically discuss the relevance of some of the differences we identified, we calculated Cohen’s d, paying more attention to larger effects at the expense of smaller ones. Cohen’s d was calculated with the free program G*Power, version 3.1.9.6.

We performed the statistical test to verify the normality of the data distribution for the items, which were further included in the analysis as dependent variables. For all data sets, the Kolmogorov–Smirnov test indicated that the participants’ responses were not normally distributed. 

## 3. Results

The aim of this study was to make a descriptive comparison between physicians who have knowingly worked with patients infected with COVID-19 and physicians who were not known to have knowingly worked with patients infected with COVID-19 in terms of their willingness to work with or without PPE, fear of losing their job if they refused to work, access to PPE, and PPE training. 

Of the 1285 physicians who participated in the study, 211 (16.4%) reported working in direct contact with patients diagnosed with COVID-19, while 1074 (83.6%) reported not working in direct contact with patients with COVID-19. When crosstabulated with gender, the batch distribution showed that these two categories of physicians were unevenly distributed by gender (χ2_(1)_ = 4.26; *p* < 0.05). For example, the frequency of males among those who reported working in direct contact with patients diagnosed with COVID-19 was significantly higher than the expected frequency of the whole group (observed count = 61, expected count = 49.4). Physicians who reported working in direct contact with patients diagnosed with COVID-19 were significantly younger (Mann–Whitney U = 98,337; Z = −0.339; *p* < 0.01), but the effect size was relatively modest (Cohen’s d = 0.23).

Table 2 presents the main socio-demographic characteristics of the sample studied.

We performed Mann–Whitney statistical tests comparing the independent groups formed by the dichotomous variable resulting from item 37f: “Healthcare professional reports having direct contact with COVID-19 patients”. Responses to this item were coded as 1 (YES) or 0 (NO). Table 3 shows the results of the Mann–Whitney comparisons. We also checked the difference between the two groups in terms of age. The age distribution is not normal, as indicated using the Kolmogorov–Smirnov test = 1.81 *p* < 0.01. The Mann–Whitney test for comparison of the two independent groups shows that physicians who reported direct contact with patients diagnosed with COVID-19 are significantly younger than those who did not report direct contact with patients diagnosed with COVID-19 (Mann–Whitney = 98,337; Z = −3.04; *p* < 0.01; Cohen’s d = 0.24).

The largest effect we found is that physicians who reported no direct contact with COVID-19 patients were significantly less willing to work with infected patients than those who reported direct contact with COVID-19 patients (U = 88,274; Z = 5.95; *p* < 0.01; d = 0.47) (Figure 2). 

Physicians who reported no direct contact with COVID-19-infected patients also reported receiving significantly less training in the use of protective equipment than those who reported direct contact with known COVID-19 patients (Mann–Whitney U = 119,354; Z = 4.91; *p* < 0.01; d = 0.40). Again, the effect size is moderate (Figure 3).

Regarding access to protective equipment, physicians who reported not being in direct contact with patients infected with COVID-19 were significantly less likely to report having access to equipment than physicians who reported being in direct contact with infected patients (Mann–Whitney U = 125.44; *p* < 0.01; Cohen’s d = 0.26). The effect size is small to moderate (Figure 4).

Another result is the response to the questions about the willingness of physicians to work when one or another item of protective equipment was missing. In each of these situations, physicians who reported that they were not in direct contact with COVID-19-infected patients were, on average, more willing to work. This was true for gloves (Mann–Whitney U = 61,164; Z = −2.82; *p* < 0.01; d = 0.23), coveralls (U = 69,199.5; Z = −1.99; *p* < 0.01; d = 0.17), k95 mask (U = 70,136; −2.12; *p* < 0.05; d = 0.16), but also for all equipment (U = 63,335; *p* < 0.01; Z = −2.78; d = 0.20). The effect sizes differed between these situations, with a larger difference between the two categories of physicians for lack of gloves (d = 0.23) and a smaller difference between the two categories of physicians for lack of masks (d = 0.16).

There were no significant differences between physicians who reported no direct contact with COVID-19-infected patients and those who reported no direct contact with COVID-19-infected patients in terms of confidence in their safety at work (U = 106,339.5; *p* > 0.05) or willingness to work given the increased risk of infecting their own family (U = 110,416.5; *p* > 0.05).

Regarding coercive motivations for working during the pandemic, physicians who reported no direct contact with COVID-19-infected patients were significantly more likely than physicians who reported direct contact with COVID-19 patients to report working during the pandemic out of fear of losing their jobs (Mann–Whitney U = 76,991; Z = −2.38; *p* < 0.05; d = 0.19) or fear of legal repercussions if they refused (Mann–Whitney U = 78,124.5; Z = −2.16; *p* < 0.05; d = 0.14). The effect is modest in both situations but somewhat larger for fear of job loss.

## 4. Discussion

The purpose of this study was to make a descriptive comparison between physicians who knowingly worked with COVID-19-infected patients and physicians who were not known to have knowingly worked with COVID-19-infected patients with regard to their willingness to work with or without PPE, fear of losing their job if they refused to work, access to PPE, and PPE training. 

Physicians who reported no direct contact with COVID-19-infected patients also reported significantly less training in the use of protective equipment than physicians who reported direct contact with known COVID-19 patients. They were significantly less likely to report having access to equipment. They were also, on average, more willing to work when one or another item of PPE was missing. The same physicians were significantly more likely to report working during the pandemic because of fear of losing their jobs or fear of legal repercussions if they refused to work. There were no significant differences in confidence in their safety at work or willingness to work in the face of increased risk of infecting their own family.

Doctors who worked directly with patients diagnosed with COVID-19 usually used the necessary protective equipment. For example, in hospital intensive care units, it was mandatory for doctors to be fully equipped. In addition, in these first-line units, the equipment was provided most of the time. During all this time, a large proportion of doctors (e.g., family doctors) were not fully equipped most of the time, because they were not necessarily considered to be working with patients with COVID-19. The interesting situation is that despite the lack of diagnosis at the time of consultation, a good proportion of patients were infected with COVID-19. For these physicians, the unrecognized risk was just as great. This idea is supported by our finding that there was no significant difference between physicians who reported no direct contact with COVID-19-infected patients and those who reported direct contact with COVID-19-infected patients in terms of confidence in their safety at work. The difference between those who knowingly and directly worked with COVID-19 cases and those who were exposed to COVID-19 cases without being diagnosed has not been addressed in literature. Our data support the hypothesis that the latter physicians were aware of the danger, and therefore, their willingness to work was lower. 

Physicians who reported no direct contact with COVID-19-infected patients were significantly less likely to report having access to equipment than physicians who reported direct contact with infected patients. The most striking similarity with the literature is that a critical factor influencing physicians’ willingness to work during the pandemic is the availability of and access to personal protective equipment (PPE). This conclusion was also explicitly stated by Hill et al. [1] and Khalid et al. [2]. Understanding of COVID-19 risks coupled with belief in the effectiveness of PPE correlated positively with willingness to work in the study by Rafi et al. [4]. Perceived vulnerability was also significantly associated with willingness to work in the study by Maraqa and coworkers [5]. In our study, we showed that physicians who reported that they did not knowingly have direct contact with patients with COVID-19 had significantly less access to personal protective equipment (PPE) and, at the same time, were less willing to work. This finding seems to be more understandable in light of both previous studies and practice. 

The effect we obtained suggests that physicians who reported no direct contact with patients diagnosed with COVID-19 may have understood the hazard at least as well as those who knowingly encountered COVID-19 patients. It begs the question why we found no difference in our study in terms of feeling safe at work, but did find a difference in willingness to work and access to PPE. The explanation may lie in the relationship between willingness to work and access to PPE. It is possible that the feeling of not being able to provide care according to ethical standards is more important than the feeling of safety itself. Also, the feeling of being respected by the institutions in which doctors work could directly affect their willingness to work, even if the perceived risk does not differ between the two categories of doctors. It is important to note that we cannot fully attribute the effect (d = 0.47) of lower willingness to work among physicians who reported no direct contact with known cases of COVID-19 to access to equipment or training received. We note that in the group we studied, physicians who reported working in direct contact with patients diagnosed with COVID-19 were significantly younger than the others. Given that some studies capture this relationship between younger age and greater willingness to work [4], it stands to reason that some of the effects we identified above may be age-related. This may be a limitation in the interpretation of the differences studied.

Physicians without direct contact with COVID-19 reported less PPE training than those with direct contact, which affected their willingness to work. Healthcare facilities with many COVID-19 cases likely prioritized training due to higher risk, while those with fewer cases or physicians not treating infected patients may have de-prioritized it due to perceived lower risk. Early in the pandemic, resource constraints may have focused training on frontline workers. Physicians with less contact with COVID-19 may not have seen immediate relevance in PPE training, especially when PPE was in short supply. Hospital administrators may have focused training on high-risk areas, potentially neglecting other departments. As knowledge of the virus increased, training likely expanded, but those not initially exposed to COVID-19 may have missed early training sessions. There may be a feedback loop in which trained physicians are more willing to work with COVID-19 patients, thus gaining more experience and training. Some hospitals may not have anticipated the need for widespread training, resulting in delays in training dissemination. In areas with lower infection rates, a false sense of security may lead to complacency in training those not directly exposed to the virus. 

Physicians not in direct contact with COVID-19 were more willing to work without certain PPE, such as gloves, overalls, and masks. Specifically, the willingness to work without gloves had a larger effect size (d = 0.23) compared to masks (d = 0.16). These physicians, with less access to and training in PPE, may not fully understand the role of PPE, leading to a greater willingness to work without it despite having a generally lower overall willingness to work. In contrast, those in direct contact with COVID-19, were more likely to value safety and PPE. The results correlate with those of Khalid et al. [2], where confidence in masks was higher (35.5%) than in gloves (21.9%). This suggests that familiarity with PPE may influence perceptions of its effectiveness. Physicians in COVID units, with rigorous training and first-hand experience, value PPE highly and are less willing to compromise. In contrast, those in less exposed roles may underestimate the risks of not using PPE. The dynamics of PPE use among physicians are influenced by training, work environment characteristics, and professional norms. For example, non-COVID settings may have different perceived hazards and unofficial standards for PPE use that are shaped by peer practices. Institutional protocols also vary, affecting PPE policies and willingness to use full PPE in different units. Physicians’ historical interactions with PPE and external influences such as media narratives also shape their perceptions and prioritization of protective elements. 

Physicians without direct contact with COVID-19 were more likely to work during the pandemic due to fear of job loss (d = 0.19) or legal repercussions (d = 0.14), with a modest but larger effect for fear of job loss. In contrast, physicians in direct contact with COVID-19 patients may feel more job security due to their frontline role. These non-contact physicians, who may have been in less critical specialties during the peak of the pandemic, felt more vulnerable to job loss. Communication from administrators may have been more coercive for them, emphasizing the consequences of noncompliance. Physicians treating COVID-19 patients may have had intrinsic motivation driven by patient care, making them less influenced by external pressures. Conversely, those not directly exposed may be more influenced by external pressures due to a less immediate health risk. Role valorization is another factor; direct patient care roles during a pandemic carry societal esteem, reducing the impact of external pressures. In contrast, those with less direct interaction may not experience this valorization, making external pressures more significant. 

The primary bias of the study arose from voluntary participation, with anonymity only partially reducing self-selection bias. Overwhelmed or dissatisfied physicians may have participated less. Distribution of the survey through professional circles could introduce another bias, favoring responses from digitally active or professionally involved physicians and underrepresenting less connected physicians. The study could not pool responses from different questions to derive cumulative scores, such as for agency or sense of duty, because of the varying applicability and diversity of situations among physicians. Despite a commendable sample size, limitations include modest effect sizes, particularly for overall willingness to work (d = 0.47) and access to PPE (d = 0.40), suggesting overall trends rather than definitive differences. Smaller effect sizes (ranging from 0.14 to 0.23) highlight the need for more extensive research.

The study did not rule out the influence of unaddressed confounding variables, so broad generalizations should be made with caution. The use of single-item measures and non-parametric statistics may have reduced the power to detect nuanced differences. Future research using complex measures, as suggested by Mushtaque et al. [3], may provide deeper insights. The survey’s “not applicable” option, while practical, may compromise statistical power and representativeness. Self-selection bias remains an issue, with potential non-representation of overworked or dissatisfied physicians and skewed distribution due to engagement with professional networks and internet accessibility. The focus on Romanian physicians limits global generalizability, as cultural differences may influence perceptions of work engagement. Global replications with culturally adapted instruments are essential for a comprehensive framework of physician willingness to work during pandemics. Strengths of the study include its large size and analytic methodology that reduces false positives.

## 5. Conclusions

Physicians who reported no direct contact with COVID-19-infected patients also reported significantly less training in the use of protective equipment than physicians who reported direct contact with known COVID-19 patients. They were significantly less likely to report having access to equipment. They were also, on average, more willing to work when one or another item of PPE was missing. The same physicians were significantly more likely to report working during the pandemic because of fear of losing their jobs or fear of legal repercussions if they refused to work. There were no significant differences in confidence in their safety at work or willingness to work in the face of increased risk of infecting their own family. We argue that despite the lack of diagnosis at the time of consultation, a large proportion of patients were, in fact, infected with COVID-19. For the physicians treating them, the unrecognized danger was equally great. The literature has been generous about the willingness of health professionals to work in general but has not addressed this difference between those who knowingly and directly worked with COVID-19 cases and those who were exposed to undiagnosed COVID-19 cases. We attempt to explain the differences we observed with other findings in the literature and some speculation derived from the authors’ own experience. 

## Figures and Tables

**Figure 1 healthcare-12-00161-f001:**
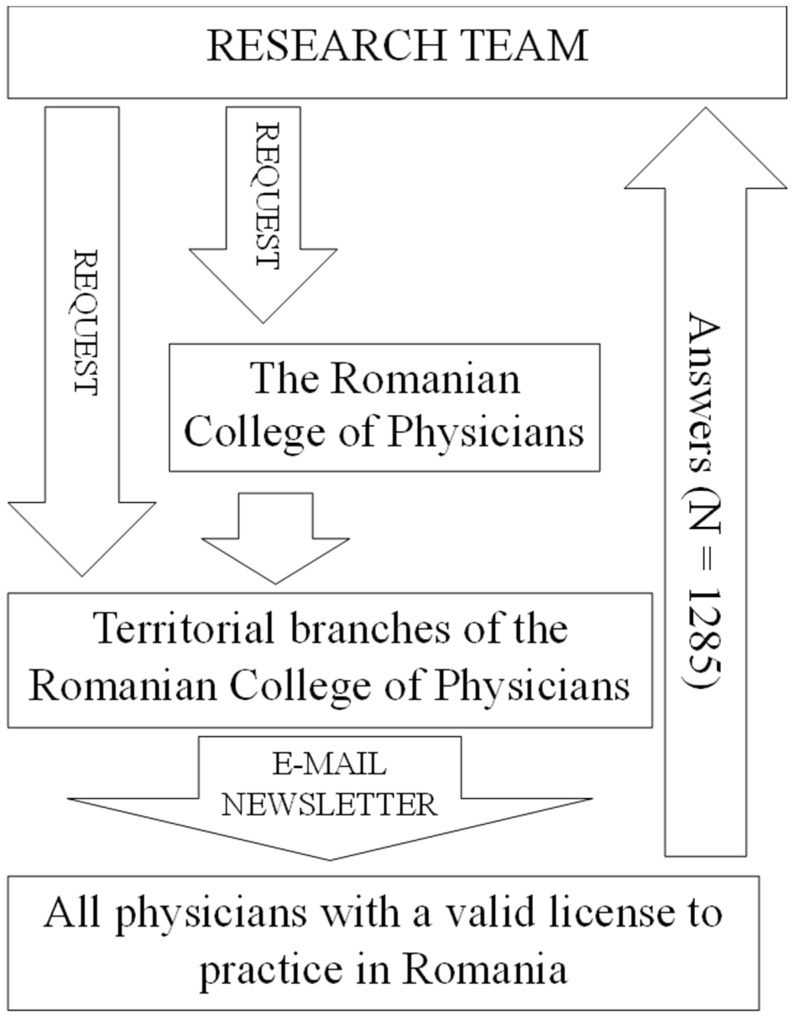
Recruitment of participants in our study by email, with the help of the Romanian College of Physicians center and territorial branches.

**Figure 2 healthcare-12-00161-f002:**
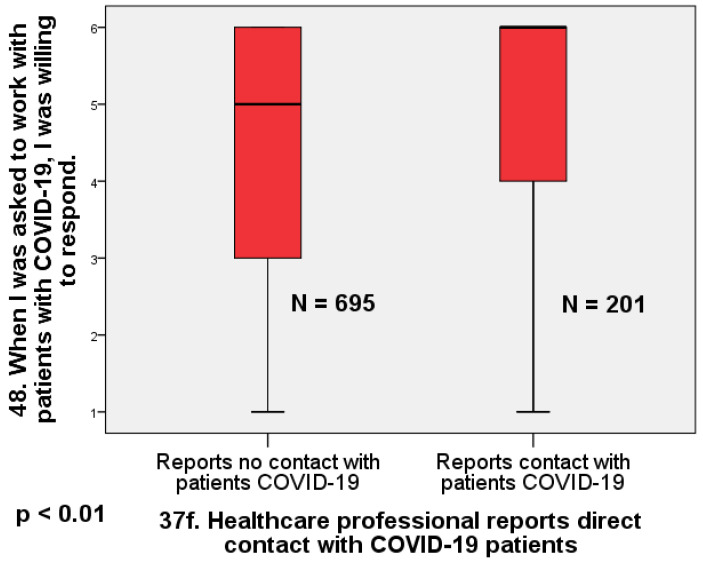
Box plot comparing doctors who reported direct contact with infected patients and those who did not report direct contact in terms of willingness to work (Mann–Whitney U = 88,274; Z = 5.95; *p* < 0.01; d = 0.47).

**Figure 3 healthcare-12-00161-f003:**
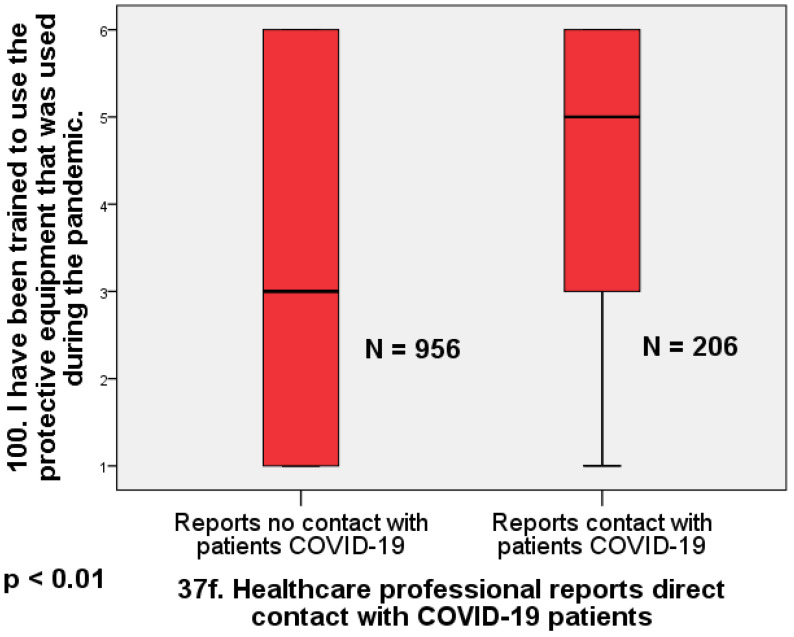
Box plot comparing doctors who reported direct contact with infected patients and those who did not report direct contact in terms of training in the use of protective measures (Mann–Whitney U = 119,354; Z = 4.91; *p* < 0.01; d = 0.40).

**Figure 4 healthcare-12-00161-f004:**
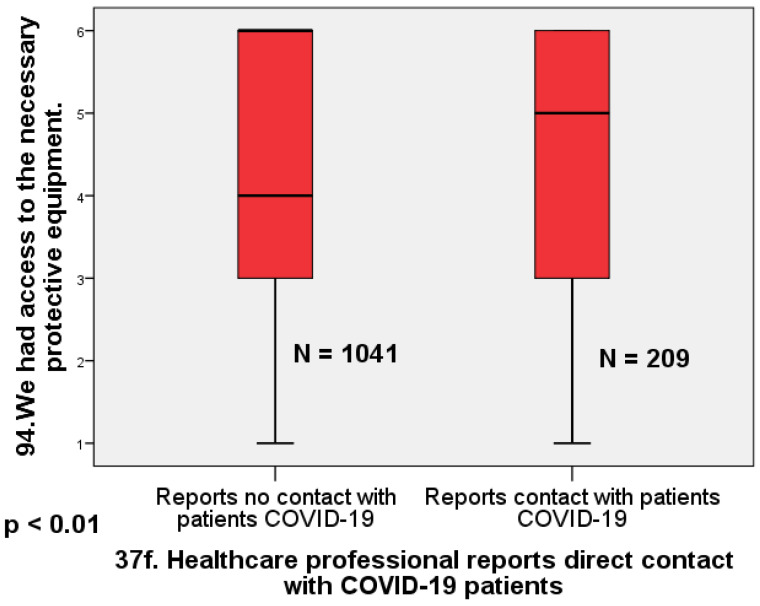
Box plot comparing doctors who reported direct contact with infected patients and those who did not report direct contact in terms of reported access to protective equipment (Mann–Whitney U = 125.44; *p* < 0.01; Cohen’s d = 0.26).

**Table 1 healthcare-12-00161-t001:** The main items used in our study, including variable types and details of response collection.

Item No.	Variable Description	Type of Measure	Observations
01	Participant’s sex	Dichotomic	
02	Age of healthcare professional in years	Scale	
03	Romania’s regional healthcare works in	Categorial	Nine historical regions
04	Type of community	Categorial	Four categories from village to large city
05	Marital status	Categorial	Married, unmarried, relationship, and other
06	How many people do you currently live with?	Scale	
07	Do you currently live with a family member?	Dichotomic	
08	How many children/youth between the ages of 6 and 18 live with you?	Scale	
09	Number of children under the age of 5 living with you?	Scale	
10	How many people over the age of 65 currently live with you?	Scale	
11	How many rooms do you have in your home?	Scale	
12	Is there any other source of income in your household besides your healthcare professional’s salary?	Dichotomic	
13	On a scale of 1 to 10, how would you rate your economic status?	Scale	1—much below average 10—much above average
14	Are you currently employed?	Dichotomic	
15	Did you lose your job as a result of the COVID-19 pandemic?	Dichotomic	
16	Do you have a source of income?	Dichotomic	
17	How much have you suffered economically since the pandemic?	Scale	1—no loss at all 10—major losses
18	What type of work do you do?	Categorial	Full-time, part-time, or retired
19 a–ai	Health worker self-reported conditions	Dichotomic	For each disease report, yes/no
20	Over the past two weeks, how would you rate your physical health?	Scale	1—Very bad 2—Excellent
21	But in the last two weeks BEFORE the COVID-19 pandemic?	Scale	1—Very bad 2—Excellent
22	Your role as a healthcare professional	Categorial	Romanian system: three levels of doctors.
23	Level of education (choose the highest level of education you have completed)	Categorial	Medical school, Master’s degree, and Doctoral degree
24	Specialty of health professional	Categorial	72 categories according to the medical specialties in Romania
25	Type of practice	Categorial	Ten categories of practice according to the Romanian system
26 a–j	Where does the healthcare work?	Dichotomic	For each type of workplace, yes or no (from maternity to family medicine)
27	Years in practice		
28	On average, how many hours per week did you work during the pandemic period?	Scale	
29	Do you think it is possible to have been in the same room as a COVID-19-positive person?	Dichotomic	
30	Please indicate the exact number of confirmed cases you have been in contact with.	Scale	
31	How many of the confirmed COVID-19 patients you have worked with have died?	Scale	
32	How many times have you been tested for COVID-19 infection?	Scale	
33	Have you been hospitalized as a result of COVID-19 infection?	Scale	
34	Has anyone close to you been hospitalized as a result of COVID-19 infection?	Dichotomic	
35	Has anyone close to you died as a result of COVID-19 infection?	Dichotomic	
36	Have you worked in a COVID-19 quarantined area?	Dichotomic	
37 a–h	Types of activities reported by the healthcare professional during the pandemic.	Dichotomic	For each, yes or no (from none of the activities listed to seeing patients with respiratory symptoms)
99 a–j	List the top three factors that negatively impacted your practice during the pandemic.	Scale	Items were separated for each category. Some categories were ranked from 1 (most important) to 3 (least important).
129 a–j	List the top three sources of information during the pandemic.	Scale	Items were separated for each category. Some categories were ranked from 1 (most important) to 3 (least important).
130 a–j	List the top three sources of moral and personal support during the pandemic.	Scale	Items were separated for each category. Some categories were ranked from 1 (most important) to 3 (least important).
38–143	Questions on a Likert-type six-point scale in order to measure constructs like agency, duty, institutional trust, etc.	Scale	1 = Strongly disagree to 6 = Strongly agree

**Table 2 healthcare-12-00161-t002:** The main socio-demographic characteristics of the sample studied. Minimum, maximum, mean, and standard deviation refer to the age variable for the whole group and separately by gender.

	Males (N = 302)	Females (N = 982)	Total Sample
Mean	49.72	47.74	48.21
Standard deviation	11.82	10.37	10.75
Minimum	26	25	25
Maximum	81	86	86

**Table 3 healthcare-12-00161-t003:** Results of statistical analyses for each item used as dependent variable. The table contains the result of testing the normality of the data distribution using the Kolmogorov–Smirnov method, the result of non-parametric Mann–Whitney comparisons as well as the effect size expressed using Cohen’s d.

Item No.	Item Statement	Kolmogorov–Smirnov *	Mann–Whitney U	Mann–Whitney Z	Mann–Whitney *p*	Cohen’s d
48	When I was asked to work with patients with COVID-19, I was willing to respond.	7.14	88,274	5.95	<0.01	0.47
51	I was willing to provide direct patient care even though I didn’t have access to coveralls when I should have been using them.	7.13	69,199.5	−1.99	<0.01	0.17
52	I was willing to provide direct patient care even though I didn’t have access to gloves when I should have been using them.	9.50	61,164	−2.82	<0.01	0.23
53	I was willing to provide direct patient care even though I did not have access to the K95 mask when I should have been using it.	8.14	70,136	−2.120	<0.05	0.16
54	I was willing to provide direct patient care without access to protective equipment when I should have been using protective equipment.	11.05	63,335.5	−2.78	<0.01	0.20
57	I trusted that I would be safe at work during the pandemic.	5.24	106,339.5	−1.20	>0.05	ns
72	I worked during the COVID-19 pandemic, even though it meant a higher risk of infecting my family than the usual risk.	12.07	110,416.5	1.21	>0.05	ns
94	We had access to the necessary protective equipment.	6.29	125,453.5	3.57	<0.01	0.26
95	I worked during the pandemic for fear of losing my job.	11.42	76,991	−2.38	<0.05	0.19
96	I worked during the pandemic for fear of legal repercussions against me.	12.43	78,124.5	−2.158	<0.05	0.14
100	I have been trained to use the protective equipment that was used during the pandemic.	6.13	119,354	4.91	<0.01	0.40

* All K-S statistics are significant at *p* < 0.01.

## Data Availability

The data presented in this study are available upon request from the corresponding author.

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
