# Peer review of "Assessment of Physicians’ Willingness to Work with Patients Not Yet Diagnosed with COVID-19 in a Romanian Sample"

_healthcare, 2024, doi:10.3390/healthcare12020161_

Round 1

Reviewer 1 Report

Comments and Suggestions for Authors

Dear authors,

The COVID-19 pandemic had forced the reorganization of healthcare systems, impacting various aspects, including hospital resources, medical staff, equipment, and supplies. Respiratory equipment, personal protective equipment, and medications were reallocated to areas with the highest demand.

Your work addresses workforce willingness to work, a crucial element of healthcare delivery, especially in the context of the COVID-19 pandemic. This is of significant importance due to its potential impact on healthcare systems and worker safety. I believe that this topic is suitable for submission to this journal. Next, I would like to provide some considerations about the structure and content of your article in its various sections.

Abstract

Please consider revising the abstract. In this section, you initiated a description of the statistical approach, but there is a lack of information regarding the data collection instruments and a minimal characterization of the sample. For instance, a brief description of the sample based on gender, age, medical specialty, and the type of questionnaire used would be beneficial. Additionally, it would be helpful to include information about when the study was conducted and whether it was a single-center or multi-center study. This information should be at the title.

In my opinion, the "Key Words" should include "Physicians' Perceptions" (or others) as it accurately represents the sample discussed in this study.

Introduction

This section is too long. Please consider resuming and comparing the specific results of other studies in the discussion section. By doing so, the Introduction remains concise and captures the essential background and significance, while the Discussion section allows for a more in-depth exploration of the findings and their implications. This separation of content can enhance the overall readability and organization of the paper.

Please consider summarizing the main objective or research question in the concluding paragraph of the introduction, it's not a strict rule but is a good practice.

Methodology

- Structurally, the methodology could benefit from a division as follows: Consider organizing the methodology under the following subheadings: 'Participant Recruitment and Study Design,' 'Questionnaires,' and 'Statistics.'The key is to ensure that the structure effectively conveys the information and makes it easily understandable for your readers.

- It's not entirely clear how you recruited the participants, and whether it was a single-center or multi-center study, and you must highlight the criteria for their inclusion or exclusion. Please consider providing this information in a clear and detailed manner, possibly utilizing a flowchart for enhanced clarity.

- The description of the multiple online applied questionnaires is quite extensive. Please consider presenting this information in a table or a schematic format for better clarity and readability.- Additionally, you should work on enhancing the characterization of your all variables to infer the most suitable statistical approach.

- Please consider adding supplementary files with the organization of the questionnaire, and also provide information about the validation of the questionnaire for your population.

- In the statistics subtitle you can describe all the statistical tests of the SSPS version and the assumptions for their use.

- In order to analyze the 'Determinants of Physicians' Willingness to Work,' the application of regression models would be beneficial.

Results

One crucial aspect that seems to be lacking in your work is the characterization of the sample. For your study, it is vital to include a table that presents the demographic characteristics of your sample. This information is essential for a comprehensive understanding of the analysis you have conducted. While you have partially included this information at the end of the methodology, it would be beneficial to provide a more detailed and organized presentation of the sample characteristics.

One of the important variables that could significantly influence the 'Determinants of Physicians' Willingness to Work' is age. However, it's not clear why it was not considered in the analysis. It might be worth discussing the rationale for excluding this potentially relevant factor

Discussion

The discussion is quite lengthy and becomes somewhat fastidious To enhance readability, I recommend summarizing the discussion by focusing on the main topics. For instance, you can consider discussing the following key areas: Determinants of Physicians' Willingness to Work; Disparities in Access to Personal Protective Equipment; The Impact of PPE Availability and Training Variations in Confidence Levels Regarding PPE;  Influence of Peer Attitudes.

Conclusion

In my opinion, a section would also benefit from a more concise summary of the results.

Specific comments

- Please consider improving the boxplot images, by increasing the font size on the scales and including the number of analyzed cases (n) beneath each box. You should also present the significance of the performed test *, **.

I find the topic of 'Physicians' Willingness to Work' to be of significant importance within the realm of healthcare services. The results of this study have the potential to provide valuable insights for healthcare institutions in managing professionals' perceptions during extreme circumstances. However, I believe that the manuscript's structure and text require some revisions and there are certain aspects that need further attention for improvement.

Author Response

REVIEWER 1:

Abstract

Request: Please consider revising the abstract. In this section, you initiated a description of the statistical approach, but there is a lack of information regarding the data collection instruments and a minimal characterization of the sample. For instance, a brief description of the sample based on gender, age, medical specialty, and the type of questionnaire used would be beneficial. Additionally, it would be helpful to include information about when the study was conducted and whether it was a single-center or multi-center study. This information should be at the title.

Answer: We introduced the following phrase in the Abstract: "In the summer of 2020, a total of 1285 Romanian physicians participated in a single-center study. Participants (mean age = 48.21 years; 302 males and 982 females, all specialties) completed a series of single-item measures adapted from previous studies on work ethics and responsibility."

Request: In my opinion, the "Key Words" should include "Physicians' Perceptions" (or others) as it accurately represents the sample discussed in this study.

Answer: We introduced "physicians’ perceptions" as a keyword

Introduction

Request: This section is too long. Please consider resuming and comparing the specific results of other studies in the discussion section. By doing so, the Introduction remains concise and captures the essential background and significance, while the Discussion section allows for a more in-depth exploration of the findings and their implications. This separation of content can enhance the overall readability and organization of the paper. Please consider summarizing the main objective or research question in the concluding paragraph of the introduction, it's not a strict rule but is a good practice.

Answer: The introduction has been shortened to 546 words. We added the following paragraph at the end of the introduction to clarify the rationale for the study and how the purpose of the study fits into the literature: "All of the studies we found examined the relationship between willingness to work and various factors but did not consider that physicians who didn't knowingly work with cases diagnosed with COVID-19 were also at risk. These risks may influence physicians' willingness to work even if the patient they are working with is not explicitly diagnosed with COVID-19. The purpose of this study is to make a descriptive comparison between physicians who have knowingly worked with patients infected with COVID-19 and physicians who are not known to have knowingly worked with patients infected with COVID-19 in terms of their willingness to work with or without PPE, fear of losing their job if they refuse to work, access to PPE, and PPE training."

Methodology

Request: Structurally, the methodology could benefit from a division as follows: Consider organizing the methodology under the following subheadings: 'Participant Recruitment and Study Design,' 'Questionnaires,' and 'Statistics.'The key is to ensure that the structure effectively conveys the information and makes it easily understandable for your readers.

Answer: We have added the following sections to the Methods section:

2.1. Participant recruitment and study design

2.2. Questionnaires

2.3. Statistics

Request: It's not entirely clear how you recruited the participants, and whether it was a single-center or multi-center study, and you must highlight the criteria for their inclusion or exclusion. Please consider providing this information in a clear and detailed manner, possibly utilizing a flowchart for enhanced clarity.

Answer: In the subsection entitled "Participant recruitment and study design" we added the following paragraph: "Physicians were recruited to participate in the study through an email sent as a newsletter directly to their email address. Access to the physicians' email addresses was provided by the Romanian National College of Physicians and the territorial colleges of physicians. The newsletter announcing participation in the study was sent a total of three times to each physician. The data collection was carried out between July 1 and August 31, 2020. Regarding the inclusion criterion, all physicians who had a valid license to practice in Romania at the time of data collection could participate in the study. The Romanian College of Physicians is the national professional organization of physicians, as an institution serving the public interest, non-governmental, apolitical, and non-profit. It is the national regulatory body for all physicians in Romania. As regards the estimation of the sample size, the aim was to cover as much as possible all doctors with a valid right to practice in Romania. No calculations were made prior to the conduct of the study regarding the estimation of the required sample size. This was a single center study. Data were collected from all over the country, but the database and statistical processing were done by a single research team".

We built Flowchart no 1 for enhanced clarity

Request: The description of the multiple online applied questionnaires is quite extensive. Please consider presenting this information in a table or a schematic format for better clarity and readability.- Additionally, you should work on enhancing the characterization of your all variables to infer the most suitable statistical approach.

Answer: We have included Table 1 with details about the questionnaire and the variables included in our study.

Request: Please consider adding supplementary files with the organization of the questionnaire, and also provide information about the validation of the questionnaire for your population.

Answer: The additional files I attach contain all the variables in the database together with all the questions. In square brackets [] are the constructs we originally intended to test by total score and internal consistency calculation (Cronbach's Alpha). As explained in the manuscript, this statistical processing was not possible afterwards.

Answer: In the „Participant Selection and Study Design” subsection we added the following paragraph: "In terms of study design, we set out to measure through several items, several constructs present in the literature: a measure of sense of duty, a measure of agency, a measure of perceived moral support, and a measure of willingness to work. The initial aim of the study was to observe whether the data in our study would correlate with each other in accordance with theoretical models found in the literature (e.g., whether an increased level of agency corresponds to an increased level of willingness to work). A flaw in the study design that we noticed after data collection was the situation of items that did not apply to particular cases (e.g. in item 17 "I was willing to work, even if my life partner got sick with COVID-19" many participants answered "not applicable"). Therefore, we could not proceed with data processing according to the original design because we could not perform parametric statistics. The number of observations that had an answer other than "not applicable" to all questions in a construct remained very small. In the present study, the independent variable was the dichotomous measure: knowingly worked with cases diagnosed with COVID-19 versus not knowingly worked with cases diagnosed with COVID-19. The dependent measures were the responses to items that related to willingness to work, access to PPE and other variables relevant to the literature to date, variables presented in Table 3. The present study has a purely descriptive design".

Request: In the statistics subtitle you can describe all the statistical tests of the SSPS version and the assumptions for their use.

Answer: In the Statistics subsection, we added the following paragraph: "The statistical indicators used in the study were: Kolmogorov-Smirnov test which tests the difference between a given distribution and the theoretical normal distribution. A significant Kolmogorov-Smirnov test shows that there is a significant difference between the observed data and the normal distribution. This implies that parametric statistical tests cannot be used for those data. We also used the Chi-Square statistical test to show whether there is a difference between the observed frequency and the expected frequency along the categories made up by the meeting of two categorical variables (in our case, the gender of the research participants and whether they worked with diagnosed cases). A significant Chi-Square test shows that there is a significant difference between the observed and expected frequency across categories, which means that the sexes are not evenly dis-tributed across the two categories of doctors and this may influence the observed differences to some extent. Finally, the most important statistical indicator used was the Mann-Whitney, suitable for testing whether there is a significant difference between two independent groups in the mean response to a given question asked on a scale. When the distribution of responses to that question is not normal, the Mann-Whitney test uses a rank average. It is worth adding, in this section, that using non-parametric tests on non-normally distributed data is not only methodologically sound, but also protects against false-positive results. SPSS 20.0 was used for statistical processing".

Request: In order to analyze the 'Determinants of Physicians' Willingness to Work,' the application of regression models would be beneficial.

Answer: In the Method section, subsection Questionnaires, we added the following paragraph:  "Physicians also responded to statements using a six-point scale to indicate agreement or disagreement. This specific scale was designed to eliminate a midpoint, thus preventing neutral responses. The broader research was designed to examine responses in the context of theoretical relationships among self-confidence, work ethic, and professional commitment. However, data characteristics prevented linear regression due to the skewed distribution and prevalence of "not applicable" options. These options were critical to reflect realistic scenarios. An example is item 48: "When asked to work with COVID-19 patients, I was willing to respond. Because not all physicians were asked to work with COVID-19 patients, there was a "not applicable" response option. Another example is item 53: "I was willing to provide direct patient care even though I did not have access to a K95 mask, although I should have used it. Many physicians have had access to masks from the be-ginning and have never been faced with this situation where they had to decide whether to risk their health. Given these data limitations, nonparametric treatment of the data was deemed the most appropriate analytical approach. It is also important to specify that a logistic regression model would have been appropriate, especially since linear regression could not be used for the reasons stated above. However, the logic of the design would have required that the dichotomous variable used as a criterion in the logistic regression approach represented a temporally ordered phenomenon according to the aspects stated in the predictors. Since it cannot be claimed that factors related to training and use of PPE in any way predict the membership of doctors in the category of those who worked with explicitly diagnosed cases or not, it would have been illogical to use a regression model".

Answer: In the Method section, subsection Questionnaires, we added the following paragraph:  " In terms of validating the questionnaires on the study population, the items we included in the study were meant to be part of some constructs (some factors or dimensions) such as sense of duty, perception of a threat, sense of trust in the institution, sense of trust in the government etc. Because most of the items had a "not applicable" variant that we had to introduce, we had to forego aggregating the items into constructs (in separate questionnaires), calculating Cronbach's Alpha internal consistency index and calculating a total score on the scales. This unfortunately resulted in an existing collection of single-item measures. However, where we were able to identify a criterion variable, we conducted binomial logistic regressions which we have published elsewhere. For the present study, we can present a comparative analysis of the two types of physicians: those who knowingly worked with cases diagnosed with COVID-19 and those who did not knowingly work with such cases ".

Results

Request: One crucial aspect that seems to be lacking in your work is the characterization of the sample. For your study, it is vital to include a table that presents the demographic characteristics of your sample. This information is essential for a comprehensive understanding of the analysis you have conducted. While you have partially included this information at the end of the methodology, it would be beneficial to provide a more detailed and organized presentation of the sample characteristics.

Answer: We have included Table 2 with the main socio-demographic characteristics of the sample studied.

Request: One of the important variables that could significantly influence the 'Determinants of Physicians' Willingness to Work' is age. However, it's not clear why it was not considered in the analysis. It might be worth discussing the rationale for excluding this potentially relevant factor

Answer: ”We included in the Results section the following paragraph: We also checked the difference between the two groups in terms of age. The age distribution is not normal, as indicated by the Kolmogorov-Smirnov test = 1.81 p <.01. The Mann Whitney test for comparison of the two independent groups shows that physicians who reported direct contact with patients diagnosed with COVID-19 are significantly younger than those who did not report direct contact with patients diagnosed with COVID-19 (Mann Whitney = 98337; Z = -3.04; p < .01; Cohen's d = .24)”.

Answer: In the Discussion section, this paragraph is present in the initial manuscript: ”It is important to note that we cannot attribute the full effect (d = 0.47) of lower will-ingness to work among physicians who reported not having direct contact with known cases of COVID-19 to access to equipment or training received. We note that in the group we studied, physicians who reported working in direct contact with patients diagnosed with COVID-19 were significantly younger than the others. Given that some studies cap-ture this relationship between younger age and greater willingness to work [4], it stands to reason that some of the effect we identified above may be age-related. This may be a limi-tation in the interpretation of the difference studied”.

Discussion

Request: The discussion is quite lengthy and becomes somewhat fastidious To enhance readability, I recommend summarizing the discussion by focusing on the main topics. For instance, you can consider discussing the following key areas: Determinants of Physicians' Willingness to Work; Disparities in Access to Personal Protective Equipment; The Impact of PPE Availability and Training Variations in Confidence Levels Regarding PPE;  Influence of Peer Attitudes.

Answer: The discussion section has been significantly reduced, but we kept the original structure because it makes more sense to us.

Conclusion

Request: In my opinion, a section would also benefit from a more concise summary of the results.

Answer: The conclusion section has been significantly reduced.

Specific comments

Request: Please consider improving the boxplot images, by increasing the font size on the scales and including the number of analyzed cases (n) beneath each box. You should also present the significance of the performed test *, **.

Answer: All images have been modified according to the specifications above.

Reviewer 2 Report

Comments and Suggestions for Authors

This manuscript addresses  results of a questionnaire with 1285 physicians who worked with patients without confirmed  covid-19 testing and their exposition to risks. In introduction, there are researches from different groups that show the use of adequate personal protective equipment as a significant factor in determining the willingness to work safely.

Methods are well explained, in details, and questions divided in different topics related to socio-demographic elements, speciality and health care.

Results are clearly demonstrated e conclusions are adequate. I just would suggest, in possible, to prepar a table with the results and the questionnaire.

Author Response

Request: I just would suggest, in possible, to prepare a table with the results and the questionnaire.

Answer: The additional files I attach contain all the variables in the database together with all the questions. In square brackets [] are the constructs we originally intended to test by total score and internal consistency calculation (Cronbach's Alpha). As explained in the manuscript, this statistical processing was not possible afterwards.

Answer: In terms of a table of results, Table 3 in the manuscript provides details only of the questions covered in this study, approached as single-item measures.

Reviewer 3 Report

Comments and Suggestions for Authors

Overall, the study is of low quality, soundness, and significance to the general audience. I briefly explain the reasons for this rating. 

The study is lengthy.

For an observational study, a 16-page manuscript is unnecessary.

The introduction is more than 2,000 words in length. This part of the manuscript seems more like a review than a scientific effort to put the context for the question addressed in the study. Additionally, I found that the authors made some plagiarism of the studies they cited because they presented a complete paragraph between 400-500 words from a single reference. The authors presented an excellent overview and summary of the previous studies published, but this is of course not the aim of the present study. The entire introduction should be rewritten and limited to present a brief context where to land their question and objective. In its actual form, seems more like an informative effort to understand the importance of other authors' work. 

The authors should reduce the introduction to 400-600 words and include lines 191-196 for the rationale of their study.

The methods are deficient. These should be presented with headings and subheadings to separate the sections and methodological framework. There is no study design, sample size estimation, definitions, and other important components to understand and replicate the study. The authors mention the questions, but they do not present them. 

There is no "statistical analysis" section within the manuscript. This is a major flaw that denotes a poor understanding of what was done and a deficient preparation of the manuscript.

The discussion is >3,500 words in length, which in some journals may be near to the length of a full-text article. This is an abuse from the authors. 

Author Response

Request: The study is lengthy. For an observational study, a 16-page manuscript is unnecessary.

Answer: The total length of the entire manuscript has been significantly reduced.

Request: The introduction is more than 2,000 words in length. This part of the manuscript seems more like a review than a scientific effort to put the context for the question addressed in the study. The authors presented an excellent overview and summary of the previous studies published, but this is of course not the aim of the present study. The entire introduction should be rewritten and limited to present a brief context where to land their question and objective. In its actual form, seems more like an informative effort to understand the importance of other authors' work. The authors should reduce the introduction to 400-600 words and include lines 191-196 for the rationale of their study.

Answer: The introduction has been shortened to 546 words. We added the following paragraph at the end of the introduction to clarify the rationale for the study and how the purpose of the study fits into the literature: "All of the studies we found examined the relationship between willingness to work and various factors but did not consider that physicians who didn't knowingly work with cases diagnosed with COVID-19 were also at risk. These risks may influence physicians' willingness to work even if the patient they are working with is not explicitly diagnosed with COVID-19. The purpose of this study is to make a descriptive comparison between physicians who have knowingly worked with patients infected with COVID-19 and physicians who are not known to have knowingly worked with patients infected with COVID-19 in terms of their willingness to work with or without PPE, fear of losing their job if they refuse to work, access to PPE, and PPE training."

Request: The methods are deficient. These should be presented with headings and subheadings to separate the sections and methodological framework. There is no study design, sample size estimation, definitions, and other important components to understand and replicate the study.

Answer: We have added the following sections to the Methods section: 2.1. Participant recruitment and study design, 2.2. Questionnaires and 2.3. Statistics

Answer: In the subsection entitled "Participant recruitment and study design" we added the following paragraph: "Physicians were recruited to participate in the study through an email sent as a newsletter directly to their email address. Access to the physicians' email addresses was provided by the Romanian National College of Physicians and the territorial colleges of physicians. The newsletter announcing participation in the study was sent a total of three times to each physician. The data collection was carried out between July 1 and August 31, 2020. Regarding the inclusion criterion, all physicians who had a valid license to practice in Romania at the time of data collection could participate in the study. The Romanian College of Physicians is the national professional organization of physicians, as an institution serving the public interest, non-governmental, apolitical and non-profit. It is the national regulatory body for all physicians in Romania. As regards the estimation of the sample size, the aim was to cover as much as possible all doctors with a valid right to practice in Romania. No calculations were made prior to the conduct of the study regard-ing the estimation of the required sample size. "

Answer: In the subsection entitled "Participant recruitment and study design" we added the following paragraph: " In terms of study design, we set out to measure through several items, several constructs present in the literature: a measure of sense of duty, a measure of agency, a measure of perceived moral support, and a measure of willingness to work. The initial aim of the study was to observe whether the data in our study would correlate with each other in accordance with theoretical models found in the literature (e.g., whether an increased level of agency corresponds to an increased level of willingness to work). A flaw in the study design that we noticed after data collection was the situation of items that did not apply to particular cases (e.g. in item 17 "I was willing to work, even if my life partner got sick with COVID-19" many participants answered "not applicable"). Therefore, we could not proceed with data processing according to the original design because we could not perform parametric statistics. The number of observations that had an answer other than "not applicable" to all questions in a construct remained very small. In the present study, the independent variable was the dichotomous measure: knowingly worked with cases diagnosed with COVID-19 versus not knowingly worked with cases diagnosed with COVID-19. The dependent measures were the responses to items that related to willingness to work, access to PPE and other variables relevant to the literature to date, variables presented in Table 1. The present study has a purely descriptive design".

Request: The authors mention the questions, but they do not present them.

Answer: The additional files I attach contain all the variables in the database together with all the questions. In square brackets [] are the constructs we originally intended to test by total score and internal consistency calculation (Cronbach's Alpha). As explained in the manuscript, this statistical processing was not possible afterwards.

Request: There is no "statistical analysis" section within the manuscript. This is a major flaw that denotes a poor understanding of what was done and a deficient preparation of the manuscript.

Answer: We separated a „Statistics” subsection and moved there all the relevant information. Other information has been added to the manuscript.

Request: The discussion is >3,500 words in length, which in some journals may be near to the length of a full-text article. This is an abuse from the authors.

Answer: We have significantly reduced the discussion section.

Round 2

Reviewer 3 Report

Comments and Suggestions for Authors

The title should be changed, remove “When the danger is not on the records” because it is not a horror movie. Include something like “Assessment of physicians’ willingness…” Also, include the country from where the study was performed to avoid generalization.

Line 34: What is PPE? This was not defined in the manuscript. 

Line 73: “The purpose of this study is to make a descriptive” This kind of statements should be written in past tense. The same goes for the Methods section, results, and discussion sections.

The methods should be written in past tense because it is mixed between present and past tense.

Figure 1 is of such low-quality for a scientific publication, it lacks clarity. The diagram should facilitate understanding how the study was performed.  

Line 184: “calculating Cronbach's Alpha internal consistency index” this should be also included in the statistical section. Besides, in this case the authors did not include an explanation for the use of this test. The authors presented their study in an heterogeneous form.

Line 200: Explain what an “Statistical indicator” is because I am not familiar with such term. I should mention that I am a statistics professor by the way. These suppose statistical indicators to which the authors refers are “Statistical test”.

Line 202-204: “A significant Kolmogorov-Smirnov test shows that there is a significant difference between the observed data and the normal distribution. This implies that parametric statistical tests cannot be used for those data.” Avoid including this kind of description because they are only used by novel scientist with little formation on scientific research. Instead, include a reference for each test. The same goes for Chi-square test and MW. 

Line 218-227: I did not understand why did the authors presented these lines in the Methods sections as these should be included in the results. Additionally, the authors suddenly introduced Cohen’s d without previous mentioning that it should be used and the purpose for its inclusion. 

Line 229: “which were further included in the analysis as dependent variables” please explain which was the independent variable. 

Line 231: “normally distributed. Table 2 presents the results of the normality test.” Actually, Table 2 presents socio-demographic data.

Table 2 Socio-demographic characteristics. Please remove this Table as it is limited to a single variable, which was not specified. Avoid using these elements that provide limited information. In addition, all statistical analysis was based on a non-parametric approach, but this table presents mean ± SD with min and max. This is a flaw.

Line 282-284: “Another result that completes the picture of the information obtained from the statistical processing is the answer to the questions related to the willingness of the doctors to work when one or another of the items constituting the protective equipment, such as the coverall, gloves or mask, was missing.” “This was true for gloves (Mann-Whitney U = 61164; Z = -2.82; p < 0.01; d = 0.23), coveralls (U = 69199.5; Z = -1.99; < .01; d = 0.17), k95 mask (U = 70136; -2.12; p < 0.05; d = 0.16), but also for all equipment (U = 63335; p < 0.01; Z =-2.78; d = 0.20). The effect sizes differed between these situations, with a larger difference between the two categories of physicians for lack of gloves (d = 0.23) and a smaller difference between the two categories of physicians for lack of mask (d = 0.16).” Please avoid being ambiguous and repetitive

According to Table 1, items 38-143 (105 items) were assessed with a Likert scale and were categorized as YES or No for Item 37f. Despite a great number of items were included in the inventory, the authors only found significant differences in Table 3 for items 48, 51, 52, 53, 54, 57, 72, 94, 95 96, and 100 (11 times out of 105). The authors did also find differences for using gloves, coveralls, K95 masks and all equipment. Similarly, fear of losing their jobs and legal repercussions differed between groups. However, except for confidence in their safety at work, willingness to work, which were not significantly different, the authors did not present the results for all other items assessed. This is considered a reporting bias and selection bias because the authors only focused on results that support their findings and excluded all other results. This is a major flaw.

Line 310-326: “We start with…therefore, their willingness to work was lower” Within this long paragraph, some sentences can be excluded because they belong to the introduction not the discussion. For instance “The literature has been generous about the willingness of health professionals to work in general, but has not addressed this difference between those who have knowingly and directly worked with COVID-19 cases and those who have been exposed to COVID-19 cases without being diagnosed.” 

After presenting the objective of the study in the first paragraph, the authors should present the most significant findings from their study. This can help to open ideas for further discussion while also indicate whether the objective was addressed or not.

Line 328-343: The following paragraph “Physicians who were not in direct contact with COVID-19 patients reported less access to PPE and were less willing to use it, which is consistent with existing literature and studies such as Hill et al [1] and Khalid et al [2]. Understanding of COVID-19 risks and 330

belief in the effectiveness of PPE, as shown by Rafi et al [4], as well as perceived vulnerability [5], influenced their willingness to work. Healthcare workers without direct contact with COVID-19 perceived similar hazards and showed no difference in safety confidence or risk of infection for their families, but their willingness to work was lower due to less access to PPE. The lack of PPE increases healthcare workers' risk of viral infection and affects their ethical dilemma between professional duty and personal/family safety. Chronic stress from feeling unprotected can lead to burnout, reducing work effectiveness. Without PPE, physicians risk becoming disease vectors, compromising patient care stand

ards, facing financial/career consequences, and losing trust in their institutions. Colleagues' decisions not to work due to a lack of PPE can influence others in the healthcare community. In addition, the physical demands of PPE can lead to exhaustion. Overall, the relationship between PPE availability and physicians' willingness to work during COVID-19 is shaped by personal, professional, psychological, and ethical factors, consistent with findings in the literature.” presents valuable information, but no result from the present study was included to compare the findings with these ideas. 

Line 352-410: Please divide this paragraph into 2-3 separate sections to enhance readability. Additionally, avoid one-sentence statements without presenting the supporting results or references. For instance “Overwhelmed or dissatisfied physicians might have participated less, indicating research limitations.” or “Peer attitudes in non-COVID treatment groups may reinforce concerns about job security and legal repercussions, amplifying these fears within that segment

Within the discussion there are several conclusions that are not linked directly to a specific result and many of them read more like speculations. The authors should be aware to avoid unnecessary speculations not supported by their findings. For instance, “This lack of training can perpetuate a cycle of reluctance to work with infected patients, resulting in ongoing training gaps” or “”

Line 425-429 of the conclusions repeats the objectives of the study, which were already presented in other sections before. These lines should be removed. The conclusions should be reduced and rewritten to enhance readability; the text is presented more like a telegram. 

Comments on the Quality of English Language

None

Author Response

  1. The title should be changed, remove “When the danger is not on the records” because it is not a horror movie. Include something like “Assessment of physicians’ willingness…” Also, include the country from where the study was performed to avoid generalization.

Answer: The title has been changed to: Assessment of Physicians' Willingness to Work with Patients Not Yet Diagnosed with COVID-19 in a Romanian sample

  1. Line 34: What is PPE? This was not defined in the manuscript.

Answer: We changed the wording to: "the availability of personal protective equipment (PPE)."

  1. Line 73: “The purpose of this study is to make a descriptive” This kind of statements should be written in past tense. The same goes for the Methods section, results, and discussion sections.

Answer: We have changed these types of phrases to past tense in the sections you mention.

  1. The methods should be written in past tense because it is mixed between present and past tense.

Answer: Thank you so much for that comment! I've reviewed the entire Method section and put all verbs in the past tense to ensure consistency.

  1. Figure 1 is of such low-quality for a scientific publication, it lacks clarity. The diagram should facilitate understanding how the study was performed.

Answer: Other evaluators asked us for a chart showing more precisely how participants were recruited. We have endeavored to render this process as well as possible through the diagram shown.

  1. Line 184: “calculating Cronbach's Alpha internal consistency index” this should be also included in the statistical section. Besides, in this case the authors did not include an explanation for the use of this test. The authors presented their study in an heterogeneous form.

Answer: Reading your comments, we realized that we were not explicit enough in our explanations. We did not use Cronbach's Alpha. Starting with line 186 of the manuscript, we have clarified this:). ”We did not calculate a Cronbach's alpha internal consistency index or a total score for the scales”.

  1. Line 200: Explain what an “Statistical indicator” is because I am not familiar with such term. I should mention that I am a statistics professor by the way. These suppose statistical indicators to which the authors refers are “Statistical test”.

Thank you for this correction. Indeed, the word "indicator" was totally inappropriate. I used, as you suggested, the expression "statistical test".

  1. Line 202-204: “A significant Kolmogorov-Smirnov test shows that there is a significant difference between the observed data and the normal distribution. This implies that parametric statistical tests cannot be used for those data.” Avoid including this kind of description because they are only used by novel scientist with little formation on scientific research. Instead, include a reference for each test. The same goes for Chi-square test and MW.

Indeed, such a detailed description of each statistical test seems superfluous. However, other evaluators have asked us to explain each statistical test individually. Please allow us to keep this section to satisfy all the requirements of the evaluation team.

  1. Line 218-227: I did not understand why did the authors presented these lines in the Methods sections as these should be included in the results. Additionally, the authors suddenly introduced Cohen’s d without previous mentioning that it should be used and the purpose for its inclusion.

Answer: We have moved the entire paragraph to the Results section. We wrote this entire paragraph in the Statistics section: "In our research we also used the statistical calculation of Cohen's d indicator for effect size. The reason for using this indicator was to have a more competent discussion of the results. Some effects were significant, but they were small. Other effects were larger and deserved more attention. In order to critically discuss the relevance of some of the differences we identified, we calculated Cohen's d, paying more attention to larger effects at the expense of smaller ones. Cohen's D was calculated with the free program G*Power, version 3.1.9.6. " (line 224)

  1. Line 229: “which were further included in the analysis as dependent variables” please explain which was the independent variable.

Answer: we specified in the manuscript, line 116: ”In the present study, the independent variable was the dichotomous measure: knowingly worked with cases diagnosed with COVID-19 versus not knowingly worked with cases diagnosed with COVID-19”.

  1. Line 231: “normally distributed. Table 2 presents the results of the normality test.” Actually, Table 2 presents socio-demographic data.

Answer: Thank you for this comment. Indeed, it is an error. I have corrected and pointed out that it is Table number 3.

  1. Table 2 Socio-demographic characteristics. Please remove this Table as it is limited to a single variable, which was not specified. Avoid using these elements that provide limited information. In addition, all statistical analysis was based on a non-parametric approach, but this table presents mean ± SD with min and max. This is a flaw.

Answer: Thank you for bringing this to our attention. Indeed, it was a major oversight. Table number 2 refers to only two variables: the quantitative variable age and the qualitative variable gender. To this end, we have added the following sentence to the table description: "Minimum, maximum, mean, and standard deviation refer to the age variable, for the whole group and separately by gender". As for the presence of table number 2, it was explicitly requested by other evaluators to describe the studied group.

  1. Line 282-284: “Another result that completes the picture of the information obtained from the statistical processing is the answer to the questions related to the willingness of the doctors to work when one or another of the items constituting the protective equipment, such as the coverall, gloves or mask, was missing.” “This was true for gloves (Mann-Whitney U = 61164; Z = -2.82; p < 0.01; d = 0.23), coveralls (U = 69199.5; Z = -1.99; p < .01; d = 0.17), k95 mask (U = 70136; -2.12; p < 0.05; d = 0.16), but also for all equipment (U = 63335; p < 0.01; Z =-2.78; d = 0.20). The effect sizes differed between these situations, with a larger difference between the two categories of physicians for lack of gloves (d = 0.23) and a smaller difference between the two categories of physicians for lack of mask (d = 0.16).” Please avoid being ambiguous and repetitive

Answer: We improved the paragraph as follows: "Another result is the response to the questions about the willingness of physicians to work when one or another item of the protective equipment was missing. In each of these situations, physicians who reported that they were not in direct contact with COVID-19-infected patients were, on average, more willing to work. This was true for gloves (Mann-Whitney U = 61164; Z = -2.82; p < 0.01; d = 0.23), coveralls (U = 69199.5; Z = -1.99; p < .01; d = 0.17), k95 mask (U = 70136; -2.12; p < 0.05; d = 0.16), but also for all equipment (U = 63335; p < 0.01; Z =-2.78; d = 0.20). The effect sizes differed between these situations, with a larger difference between the two categories of physicians for lack of gloves (d = 0.23) and a smaller difference between the two categories of physicians for lack of mask (d = 0.16) ".

  1. According to Table 1, items 38-143 (105 items) were assessed with a Likert scale and were categorized as YES or No for Item 37f. Despite a great number of items were included in the inventory, the authors only found significant differences in Table 3 for items 48, 51, 52, 53, 54, 57, 72, 94, 95 96, and 100 (11 times out of 105). The authors did also find differences for using gloves, coveralls, K95 masks and all equipment. Similarly, fear of losing their jobs and legal repercussions differed between groups. However, except for confidence in their safety at work, willingness to work, which were not significantly different, the authors did not present the results for all other items assessed. This is considered a reporting bias and selection bias because the authors only focused on results that support their findings and excluded all other results. This is a major flaw.

Answer: The research covered a large number of variables. For the present study, we selected only those variables that were related to doctors' willingness to work. We tested whether there was a significant difference between doctors who knowingly worked with patients infected with COVID and those who worked with patients without knowing whether they were infected or not, in terms of aspects relevant to this difference such as the presence of various means of protection, the threat from authorities or the feeling of safety at work. Differences were not statistically tested for the other variables. The way in which we selected the items mentioned is related to the clinical experience of the authors of this study and the literature. All the differences tested are those shown in Table number 3.

  1. Line 310-326: “We start with…therefore, their willingness to work was lower” Within this long paragraph, some sentences can be excluded because they belong to the introduction not the discussion. For instance “The literature has been generous about the willingness of health professionals to work in general, but has not addressed this difference between those who have knowingly and directly worked with COVID-19 cases and those who have been exposed to COVID-19 cases without being diagnosed.”

Answer: In this paragraph, we have only managed to delete two statements that seemed superfluous (lines 335, 347). The rest of the ideas seem relevant in this section. They connect the research idea with our findings.

  1. After presenting the objective of the study in the first paragraph, the authors should present the most significant findings from their study. This can help to open ideas for further discussion while also indicate whether the objective was addressed or not.

We inserted at the beginning of the Discussion section, the following paragraph, restating the main findings: "Physicians who reported no direct contact with COVID-19-infected patients also reported significantly less training in the use of protective equipment than physicians who reported direct contact with known COVID-19 patients. They were significantly less likely to report having access to equipment. They were also, on average, more willing to work when one or another item of PPE was missing. The same physicians were significantly more likely to report working during the pandemic because of fear of losing their jobs or fear of legal repercussions if they refused to work. There were no significant differences in confidence in their safety at work or willingness to work in the face of increased risk of infecting their own family".

  1. Line 328-343: The following paragraph “Physicians who were not in direct contact with COVID-19 patients reported less access to PPE and were less willing to use it, which is consistent with existing literature and studies such as Hill et al [1] and Khalid et al [2]. Understanding of COVID-19 risks and belief in the effectiveness of PPE, as shown by Rafi et al [4], as well as perceived vulnerability [5], influenced their willingness to work. Healthcare workers without direct contact with COVID-19 perceived similar hazards and showed no difference in safety confidence or risk of infection for their families, but their willingness to work was lower due to less access to PPE. The lack of PPE increases healthcare workers' risk of viral infection and affects their ethical dilemma between professional duty and personal/family safety. Chronic stress from feeling unprotected can lead to burnout, reducing work effectiveness. Without PPE, physicians risk becoming disease vectors, compromising patient care stand ards, facing financial/career consequences, and losing trust in their institutions. Colleagues' decisions not to work due to a lack of PPE can influence others in the healthcare community. In addition, the physical demands of PPE can lead to exhaustion. Overall, the relationship between PPE availability and physicians' willingness to work during COVID-19 is shaped by personal, professional, psychological, and ethical factors, consistent with findings in the literature.” presents valuable information, but no result from the present study was included to compare the findings with these ideas.

Answer: Thank you for your comment. Re-reading the paragraph, we realized that it was not at all clear what we wanted to convey. We have reworded the paragraph as follows: "Physicians who reported not being in direct contact with patients infected with COVID-19 were significantly less likely to report having access to equipment than physicians who reported being in direct contact with infected patients. The most striking similarity with the literature is the following: a critical factor influencing physicians' willingness to work during the pandemic is the availability of and access to personal protective equipment (PPE). This conclusion was also explicitly reached by Hill et al. [1] and Khalid et al [2]. Understanding of COVID-19 risks coupled with belief in the effectiveness of PPE correlated positively with willingness to work in the study by Rafi et al. [4]. Perceived vulnerability was also significantly associated with willingness to work in the study by Maraqa and coworkers [5]. In our study, we showed that physicians who reported that they did not knowingly have direct contact with patients with COVID-19 had significantly less access to personal protective equipment (PPE) and at the same time were less willing to work. This finding seems to be more understandable in the light of both previous studies and practice.  The effect we obtained suggests that physicians who reported that they did not come into direct contact with patients diagnosed with COVID-19 may have understood the danger at least as well as those who knowingly encountered COVID-19 patients. It begs the question why we found no difference in our study in terms of feeling safe at work, but we did find a difference in willingness to work and access to PPE.  The explanation may lie in the link between willingness to work and access to PPE. It is possible that the feeling of not being able to provide care according to ethical standards is more important than the feeling of safety itself. Also, the feeling of being respected by the institutions in which doctors work could directly affect their willingness to work, even if the perceived risk does not differ between the two categories of doctors".

  1. Line 352-410: Please divide this paragraph into 2-3 separate sections to enhance readability. Additionally, avoid one-sentence statements without presenting the supporting results or references. For instance “Overwhelmed or dissatisfied physicians might have participated less, indicating research limitations.” or “Peer attitudes in non-COVID treatment groups may reinforce concerns about job security and legal repercussions, amplifying these fears within that segment”

Answer: Indeed, the paragraph you mentioned had a formatting error. Four separate paragraphs are there, three of which relate to the results and one of which presents some of the limitations of the study. In terms of wording, some statements were indeed too confusing, and we have removed them. As for the range of possible explanations, it is true that they have no reference in the literature. The reason is that those possible explanations are based on the direct experience of the authors (some of whom play key roles in professional associations in Romania). Thank you once again for your careful attention to the coherence of the document.

  1. Within the discussion there are several conclusions that are not linked directly to a specific result and many of them read more like speculations. The authors should be aware to avoid unnecessary speculations not supported by their findings. For instance, “This lack of training can perpetuate a cycle of reluctance to work with infected patients, resulting in ongoing training gaps” or “”

Answer: We have deleted the sentence you refer to. As far as explanations are concerned, we insist that we have addressed a rather narrow issue that is not present in the literature. Therefore, many of our explanations are hypothetical, as no explanation would be found in the literature on the differences between the two categories of doctors we investigated.

  1. Line 425-429 of the conclusions repeats the objectives of the study, which were already presented in other sections before. These lines should be removed. The conclusions should be reduced and rewritten to enhance readability; the text is presented more like a telegram.

Answer: We have reduced the conclusion further: "Physicians who reported no direct contact with COVID-19-infected patients also reported significantly less training in the use of protective equipment than physicians who reported direct contact with known COVID-19 patients. They were significantly less likely to report having access to equipment. They were also, on average, more willing to work when one or another item of PPE was missing. The same physicians were significantly more likely to report working during the pandemic because of fear of losing their jobs or fear of legal repercussions if they refused to work. There were no significant differences in confidence in their safety at work or willingness to work in the face of increased risk of infecting their own family. We argue that despite the lack of diagnosis at the time of consultation, a large proportion of patients were in fact infected with COVID-19. For the physicians treating them, the unrecognized danger was just as great. The literature has been generous about the willingness of health professionals to work in general but has not addressed this difference between those who knowingly and directly worked with COVID-19 cases and those who were exposed to undiagnosed COVID-19 cases. We attempt to explain the differences we observed with other findings in the literature, but also with some speculation derived from the authors' own experience".